# Effects of Dietary Ferulic Acid Supplementation on Hepatic Injuries in Tianfu Broilers Challenged with Lipopolysaccharide

**DOI:** 10.3390/toxins14030227

**Published:** 2022-03-21

**Authors:** Gang Shu, Ziting Tang, Hong Du, Yilei Zheng, Lijen Chang, Haohuan Li, Funeng Xu, Hualin Fu, Wei Zhang, Juchun Lin

**Affiliations:** 1Department of Basic Veterinary Medicine, Sichuan Agricultural University, Chengdu 611100, China; dyysg2005@sicau.edu.cn (G.S.); tangziting67@outlook.com (Z.T.); duhong96@hotmail.com (H.D.); lihaohuan7@163.com (H.L.); funengxu@sicau.edu.cn (F.X.); fuhl.sicau@163.com (H.F.); zhangwei26510c@126.com (W.Z.); 2Department of Veterinary Medicine, University of Minnesota, Minneapolis, MN 55791, USA; zhen0219@umn.edu; 3Department of Veterinary Clinical Science, College of Veterinary Medicine, Oklahoma State University, Stillwater, OK 74078, USA; lj.chang@okstate.edu

**Keywords:** broiler, ferulic acid, hepatic injury, lipopolysaccharide

## Abstract

Lipopolysaccharide (LPS) is an endotoxin that can cause an imbalance between the oxidation and antioxidant defense systems and then induces hepatic damages. Ferulic acid (FA) has multiple biological functions including antibacterial and antioxidant activities; however, the effect of FA on lipopolysaccharide-induced hepatic injury remains unknown. The purpose of this study was to investigate the mechanism of action of dietary Ferulic acid against Lipopolysaccharide-induced hepatic injuries in Tianfu broiler chickens. The results showed that supplementation of FA in daily feed increased body weight (BW) and decreased the feed conversion ratio (FCR) in LPS treatment broilers significantly (*p* < 0.05). Additionally, supplement of FA alleviated histological changes and apoptosis of hepatocytes in LPS treatment broilers. Supplement of FA significantly decreases the activities of ROS. Interestingly, the levels of antioxidant parameters including total superoxide dismutase (T-SOD), total antioxidant capacity (T-AOC), and glutathione (GSH) in LPS group were significantly increased by the FA supplementation (*p* < 0.05). Nevertheless, administration of LPS to broilers decreased the expressions of Nrf2, NQO1, SOD, GSH-Px, CAT and Bcl-2, whereas it increased the expressions of Bax and Caspase-3 (*p* < 0.05). Moreover, the expressions of Nrf2, NQO1, SOD, CAT, Bcl-2 were significantly upregulated and Caspase-3 were significantly downregulated in the FL group when compared to LPS group (*p* < 0.05). In conclusion, supplementation of FA in daily feed improves growth performance and alleviates LPS-induced oxidative stress, histopathologic changes, and apoptosis of hepatocytes in Tianfu broilers.

## 1. Introduction

In a commercial poultry farm, broiler chickens are more likely to suffer from oxidative stress due to intensive production and abrupt environmental changes, such as formulation of diet, management of facility, pathogenic microorganisms, and so on [1]. It has been indicated that oxidative stress leads to macromolecular damages, resulting in hepatic disease, which is a leading cause of liver damage caused by various factors [2,3]. Lipopolysaccharide (LPS), the essential component of outer membrane of Gram-negative bacteria, recognized a crucial role in the development of oxidative stress [3,4,5]. Large amounts of LPS are released after bacterial death and then through the portal vein. Nevertheless, excessive accumulation of LPS induces hepatic damages [6,7], which is due to the inability of liver to clear bacteria or LPS [8].

The nuclear factor-erythroid 2-related factor 2 (Nrf2) is a member of the basic leucine zipper, which involves multiple aspects against oxidative stress, such as detoxification, regulating cell metabolism and promoting cell proliferation. Nrf2 binds to its inhibitor Kelch-like ECH-associated protein 1 (Keap1) homodimer in the cytoplasm under physiological conditions; however, when stimulated by ROS, keap1 is modified and Nrf2 is transported to the nucleus, then recognizes and interacts with the antioxidant responsive element (ARE), initiating the transcription of downstream antioxidant protective genes and phase detoxification enzymes genes [9]. Nrf2 regulates many redox reactions through the induction of key enzymes and genes [10]. Furthermore, more and more studies showed that the activation of Nrf2 pathway to induce the expression of cytoprotective genes is a potential option to prevent liver injury [11]. Both reactive oxygen species (ROS) and oxidative stress can serve as the inducer of apoptosis [12,13]. Oxidative stress results from excessive generation of ROS impair mitochondrial electron transports, then Cyt c is released from the mitochondrion into the cytosol, which further leads to caspase activation and apoptosis [14]. Cyt c binds to Apaf-1 and procaspase-9 to form an apoptosome, thus resulting in caspase-9 activation, which ultimately triggers PARP apoptosis [15,16]. In addition, B-cell lymphoma(Bcl-2) family protein, mitochondrial pro-apoptosis protein, and caspase are the key members in the apoptosis pathway [17].

In recent years, phytochemicals have drawn increasing attention for animal health due to their various biological activities. Compared with the use of antibiotics, phytochemicals have the advantage of non-resistance and are residue free [18]. Ferulic acid (FA,4-hydroxy-3-methoxycinnamic acid) is one of active ingredients of phenolic acid, extracted from the seeds and leaves of asafetida (*Ferula foetida* L.), and has been recognized as a low systemic toxicity antioxidant [19,20]. It has been reported recently that FA has multiple biological functions, such as antibacterial [21], antioxidant [22,23], anti-inflammatory [24], antithrombotic [25], anticancer, and antidiabetic [26] properties. A potential mechanism of FA as an antioxidant is that FA not only neutralizes the free radicals but also inhibit the formation of reactive oxygen species (ROS) [27]. Additionally, FA has been proven to increase the antioxidant function of superoxide dismutase (SOD). It reported that FA (1 μM) could protect HEK293 cells treated with H_2_O_2_ by reducing ROS production and the levels of CAT and SOD [28]. FA could dose-dependently promote nuclear translocation of Nrf2 and HO-1 expression, which can be understood as enhancing the antioxidant ability. A previous study result showed that FA prevented lipid peroxidation and cellular apoptosis by up-regulating the expression of heme oxygenase-1(HO-1) [29]. FA can reduce apoptosis through increasing the ratio of Bcl-2/Bax [30]. Furthermore, a recent study indicates that FA shows significant protective effects during cellular or tissue damage [31]. However, the information of the clinical effects of FA supplementation in LPS-induced hepatic injury chickens is limited.

The purpose of this study was to evaluate the clinical effects of supplementing FA in broiler diets and investigate the mechanism of action of FA in attenuating LPS-induced hepatic cellular apoptosis in broiler chickens. Hence, we hypothesized that the supplementation of FA will alleviate LPS-induced oxidative stress and hepatic cellular apoptosis in broiler chickens. 

## 2. Results

### 2.1. Growth Performance among Treatment Groups

The impactions of FA on growth performance of broilers among treatment groups were shown in Figure 1. The results showed that on day 21, the birds in LPS group showed significantly lower BW (*p* < 0.05) compared to those in the CON group. Convincingly, from d 15 to 21, we observed significantly lower ADG and ADFI (*p* < 0.05), with an increased FCR (*p* < 0.05) among birds in the LPS group compared to the CON group. However, in the FL group, the birds showed a significant increase in BW compared to the LPS group (*p* < 0.05). Furthermore, the FCR decreased significantly at 15 to 21 d (*p* < 0.05) in comparison to the FL and LPS groups. 

### 2.2. LPS-Induced Histopathological Changes of Hepatocytes 

The histopathological analysis was performed to determine the extent of hepatocytes injuries. Figure 2A showed intact complete hepatocytes without histopathological changes in the CON group. However, the LPS group showed obvious pathological changes characterized by hepatic cords fragmentation, cellular swelling, hemorrhage, central venous hyperemia, and diffuse necrosis (Figure 2B). However, hepatocytes in FA and FL groups (Figure 2C,D) showed less obvious histopathological changes compared to the LPS group.

### 2.3. Liver Antioxidant Status

The levels of concentration of T-SOD, T-AOC, GSH and ROS in the liver were shown in Figure 3. It showed that the levels of T-SOD, T-AOC and GSH decreased significantly (*p* < 0.05), whereas the ROS level was significantly increased (*p* < 0.05) in the LPS group when compared to the CON group. In the FL group, however, the levels of T-SOD, T-AOC and GSH elevated significantly when compared to the LPS group (*p* < 0.05). The flow cytometry quadrant diagrams of ROS were shown in Appendix A.

### 2.4. LPS-Induced Apoptosis and Changes of Mitochondrial Depolarization Ratio

Figure 4 showed the results of LPS-induced apoptosis and changes in the mitochondrial depolarization ratio. The results showed that hepatocytes apoptosis (Figure 4A) and mitochondrial depolarization ratio (Figure 4B) were significantly high in the LPS group among treatment groups (*p* < 0.05). However, the apoptosis and mitochondrial depolarization ratio decreased (*p* < 0.05) in the FA group when compared to the LPS and FL groups. The flow cytometry quadrant diagrams of apoptosis and mitochondrial depolarization ratio were shown in Appendix A, respectively.

### 2.5. Expressions of Antioxidant and Apoptosis Associated Genes

The expressions of antioxidant associated (Nrf2, NQO1, SOD, GPx, CAT) and apoptosis-associated genes (Bax, Bcl-2, Caspase-3) were shown in Figure 5. The expressions of Nrf2, NQO1, SOD, GSH-Px, CAT and Bcl-2 were downregulated significantly in the LPS group compared to the control group, whereas the expressions of Bax and Caspase-3 were upregulated (*p* < 0.05). However, in comparison to the LPS group, the expressions of Nrf2, NQO1, SOD, CAT and Bcl-2 increased and the expressions of Caspase-3 decreased significantly in the FL group (*p* < 0.05).

## 3. Discussion

The growth performance of chickens was inhibited by the administration of LPS [32,33], which is consistent with the results obtained in the current study. The decrease in the growth performance can be explained by the hypothalamic–pituitary–adrenal-axis-associated antioxidant process, which induced anorexia [34]. It has been reported by Peña Torres that the supplementation of FA increases ADG in heifers [35]. Similarly, Yu found that the supplementation of FA in daily feed with oxidized fish oil showed a positive effect in farmed tilapia due to the antioxidative potential of FA [36]. Moreover, the results obtained in this study indicate that birds fed with FA had greater BW and lower FCR in LPS-induced chickens than those fed with a basal diet.

The current study reveals that administration of LPS induces histological changes of hepatocytes with the characteristics of necrotic, cloudy swelling, and infiltration of inflammatory cells, which are similar to those demonstrated in a previous study [37]. The results of the current study prove that supplementation of FA alleviates the LPS-induced histological changes of hepatocytes in chickens. Similarly, it has been proven by other studies that supplementation of FA decreases the risk of LPS-induced hepatocytes degeneration and apoptosis [38,39]. FA shows its superior antioxidation and anti-inflammatory effects in the present and a previous study [40], although the detail mechanism of action warrants further investigation.

LPS induces the breakdown of oxidation reduction (redox) homeostasis, contributing to cellular and tissue oxidative injuries, due to ROS accumulation [41,42]. Antioxidant enzymes (SOD, GSH-Px, CAT and so on) and free radical scavenging antioxidants (vitamin E, vitamin C, GSH and so on) are responsible for protecting tissues from oxidation damage and scavenging of free radicals [43,44]. Therefore, the levels of these antioxidants in the systemic circulation are considered sensitive indicators for determining oxidative stress. The present results reveal that LPS increases the accumulation of ROS and decreases the activity of antioxidation enzymes significantly, including T-SOD, T-AOC, and GSH. However, the supplementation of FA reverses the effects caused by LPS by exhibiting the protective effects on the LPS-induced hepatocytes pathological changes, as well as increasing the levels of T-SOD, T-AOC, and GSH. The current results are in agreement with the results of [45,46]. Nrf2 is a redox-sensitive nuclear factor, regulating the expression of genes involved in the antioxidation process [47]. The inactivation of Nrf2 is induced by its inhibitor Kelch ECH-associating protein 1 (Keap1). However, at the occurrence of oxidative stress, Nrf2 is activated and released from Keap1, moving from the cytoplasm to the nucleus. Then, Nrf2 turns to regulate the expressions of downstream antioxidants such as HO-1, NQO1, SOD, GSH-Px, and CAT [48,49]. Some studies showed that FA regulates the expression of Nrf2 in the nucleus and protects neurons against ROS-associated oxidative stress [50]. In this study, the expressions of Nrf2, NQO1, SOD, GSH-Px, and CAT were decreased significantly by LPS. However, the supplementation of FA promotes the expressions of those aforementioned genes, indicating that FA inhibits LPS-induced hepatic oxidative stress via the Nrf2 signaling pathway. This finding revealed new evidence of a mechanism of action of FA against LPS-induced hepatocytes oxidative injuries in broiler chickens. 

One of the important findings of the current study is that the supplementation of FA decreases LPS-induced hepatocytes apoptosis and the mitochondrial depolarization ratio. It has been reported that accumulation of ROS and deletion of antioxidants increase the mitochondrial membrane depolarization ratio, leading to apoptosis [51]. The Bcl-2 family contains pro-apoptotic and antiapoptotic proteins, controlling the release of Cyt-C from the mitochondria to the cytosol to regulate the apoptotic signaling pathway [52]. Cyt-C activates the caspase pathway, leading to the expression of caspase-3, which is the most important executioner caspase for triggering of apoptosis [53,54]. In this study, the expression of Bax and caspase-3 increased significantly, whereas the expression of Bcl-2 decreased significantly in LPS group, which indicates an occurrence of hepatocytes apoptosis. Nevertheless, the supplementation of FA reverses LPS-induced hepatocytes apoptosis by regulating the expressions of caspase-3 and Bcl-2 in chickens.

## 4. Conclusions

In conclusion, supplementation of FA in daily feed not only improves growth performance and attenuates LPS-induced hepatocytes apoptosis, but also protects hepatocytes from oxidative stress damage and regulates the expressions of antioxidation genes positively in Tianfu broiler chickens. This study shows firsthand evidence of the mechanism of action of FA against LPS-induced hepatocytes apoptosis in Tianfu broiler chickens.

## 5. Materials and Methods

### 5.1. Reagents

Ferulic acid (HPLC = 99.91%) was purchased from Chengdu Herbpurify Co. Ltd. (Chengdu, China). *Escherichia coli* lipopolysaccharide (LPS, O55:B5; L2880) was purchased from Sigma-Aldrich (L2880). *Escherichia coli* lipopolysaccharide was reconstituted by sterile saline to 100 μg/mL. Total reactive oxygen species (ROS) assay kit was purchased from ThermoFisher Scientific Company (Waltham, MA, USA). RNAiso Plus, PrimeScript^TM^ RT reagent Kit with gDNA Eraser as well as TB Green^TM^ Premix Ex Taq^TM^ II were purchased from TaKaRa Bio Inc. (Shiga, Japan). Chicken-specific ELISA assay kits for total antioxidant capacity (T-AOC), total superoxide dismutase (T-SOD) and glutathione (GSH) were purchased from Nanjing Jiancheng Bioengineering Institute (Nanjing, China).

### 5.2. Experimental Animals and Management

This study was approved by Sichuan Agricultural University’s Institutional Animal Care and Use Committee (DYY-2018203007). The experiment was performed in the Poultry Farm of Sichuan Agricultural University and all procedures followed were in accordance with the guidelines of National Standard Laboratory Animal Requirements of Environment and Housing Facilities (GB 14925-2001). 

A total of 160 Tianfu broiler chickens (25 days old, male) were purchased from and raised in the Poultry Breeding Research Unit of Sichuan Agricultural University (Ya’an, China). The birds were provided with free food and water ad libitum. The lighting program was set to produce 16 h of light and 8 h of darkness throughout the experimental period. The ambient temperature and relative humidity were maintained at 28 ± 2 °C and 50–55%, respectively. The composition and nutrient level of the basal diet are presented in Table 1, and nutrition requirement of broiler chickens was in accordance with the guidelines of the National Research Council requirements for chickens.

### 5.3. Study Design

A total of 160 Tianfu birds with an initial body weight of 184.33 ± 3.34 g were randomly divided into four treatments, 4 replicates for each treatment and 10 chicks for each replicate: control group (CON), LPS group (LPS), FA group (FA), and FL group (FA + LPS). Birds in CON and LPS groups were fed with basal diets, whereas those in FA and FL groups were fed with basal diets and 100 mg/kg of FA mixture. On days 14, 16, 18 and 20, birds in LPS and FL group were administered with 1.0 mg/kg LPS of body weight intraperitoneally and those in CON and FA groups were administered with equal volume of normal saline intraperitoneally according to protocols described in a pervious study [55].

### 5.4. Growth Performance Parameters

Throughout the experimental period, body weight (BW) of chickens was measured on day 1, 7, 14 and 21, whereas feed intake was recorded at day 7, 14 and 21; thereafter, average daily weight gain (ADG), average daily feed intake (ADFI) and feed conversion ratio (FCR) were calculated.

### 5.5. Sample Collection and Measurements

On day 21, twelve birds per group were randomly selected and euthanized by administration of 100 mg/kg of pentobarbital sodium intravenously. Liver tissues were collected and stored at −80 °C for subsequent analysis. Liver samples were sliced and fixed in 4% paraformaldehyde. Parts of the liver samples were centrifuged and suspended in ice cold PBS (pH 7.4) to obtain 1 × 10^6^ cell/mL solution subsequent flow cytometry analysis. 

### 5.6. Histopathological Observation

The liver samples were fixed in 4% (*w*/*v*) buffered paraformaldehyde for 24 h. The trimmed samples were dehydrated by alcohol, cleared by xylene and then embedded in paraffin. The samples were sliced into 5 μm serial sections and stained by hematoxylin-eosin (HE). Histopathological changes were observed and recorded by a microscope imaging system (DM1000, Leica, Germany).

### 5.7. Activities of Antioxidants Parameters

One gram of liver sample was homogenized with normal saline at 4 °C and centrifuged at 3000× *g* for 10 min at 4 °C to obtain the supernatant. The level of T-AOC was measured using Ferric-ion-reducing antioxidant power (FRAP). The activity of T-SOD was measured by hydroxylamine method. The level of GSH was measured using the protocol that GSH react with dithiodinitrobenzoic acid (DTNB) to produce yellow compounds in the test kit.

### 5.8. Detection of Reactive Oxygen Species (ROS) and Apoptosis

Flow cytometry was used for detecting ROS level, mitochondrial membrane potential (MMP) and the percentage of apoptosis. The liver samples were crushed, filtered through a 350-mesh nylon membrane, centrifuged at 600× *g* for 5 min, and adjusted to a cell density of 1.0 × 10^6^ cell/mL with phosphate-buffered saline (PBS). To examine intracellular ROS generation, 300 μL of cell suspension was incubated with 1 μg DCFH-DA (total ROS assay kit, 10 μM, H_2_O_2_ as the positive control) at 37 °C for 15 min in the dark, then washed with cold PBS and centrifuged at for 5 min. After discarding the supernatant, the cells were resuspended in 0.4 mL PSB and counted with a CytoFLEX flow cytometer (Backman Coulter, Brea, CA, USA). To examine MMP, 100 μL suspension was incubated with 50 μL JC-1 (BD^TM^ MitoScreen) at 37 °C for 15 min in the dark, then washed with cold PBS and centrifuged at for 5 min. After discarding the supernatant, the cells were resuspended in 0.4 mL PSB and counted with a CytoFLEX flow cytometer (Backman, Brea, CA, USA). Apoptosis was analyzed using the Annexin V-FITC/PI Apoptosis Detection Kit (Invitrogen, Carlsbad, CA, USA). Briefly, 100 μL of cell suspension was stained with Annexin V-FITC and PI solution for 10 min at room temperature without exposure for 15 min The results of ROS level and MMP were presented as percentage (%) and mitochondrial depolarization ratio, respectively. All recorded data were analyzed by Kaluza 2.1 software (Beckman Coulter, Brea, CA, USA).

### 5.9. Real-Time Quantification PCR (RT-qPCR)

The samples were treated with frozen (diethylpyrocarbonate) DEPC water and stored at −80 °C until further analysis. A total of 60 mg of treated sample was homogenized by cooling tissue homogenizer. Total RNA was extracted from the tissue samples using RNAiso Plus. The concentration of RNA was measured by a NanoDrop-2000 spectrophotometer (ThermoFisher Scientific Co., Waltham, MA, USA) which was able to detect reversed transcription of 1.8–2.0 with a detecting range of 260–280 nm. A Real-Time PCR Detection Platform was used to perform RT-qPCR with a SYBR Premix Ex Taq II kit. The primer sequences for the target genes and β-actin (as housekeeping gene) were shown in Table 2. CON group was selected as a reference group in this study (value = 1), and gene expressions were quantified by 2^−∆∆Ct^ method (∆Ct = Ct _target gene_ − Ct _housekeeping gene_; ∆∆Ct = ∆Ct − ∆Ct _reference_). 

### 5.10. Statistical Analyses

All data were analyzed by SPSS 18.0 (SPSS Inc., Chicago, IL, USA). One-way analysis of variance (ANOVA) was used to determine the overall difference among all groups. All parameters were presented as mean ± standard error (mean ± SE). Values of *p* < 0.05 were determined as significant.

## Figures and Tables

**Figure 1 toxins-14-00227-f001:**
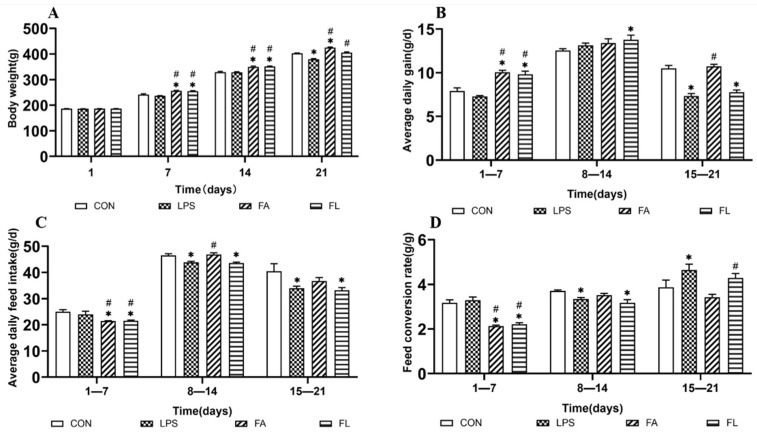
Effects of ferulic acid on growth performance in LPS-induced Tianfu broiler. (**A**) body weight (BW, g), (**B**) average daily gain = weight gain/day (ADG, g/d), (**C**) average daily feed intake = total feed intake/day (ADFI, g/d), and (**D**) feed conversion rate (FCR, g/g) of Tianfu broiler chickens in four groups. Values were shown in mean ± SE. * *p* < 0.05 compared with the CON group. # *p* < 0.05 compared with the LPS group.

**Figure 2 toxins-14-00227-f002:**
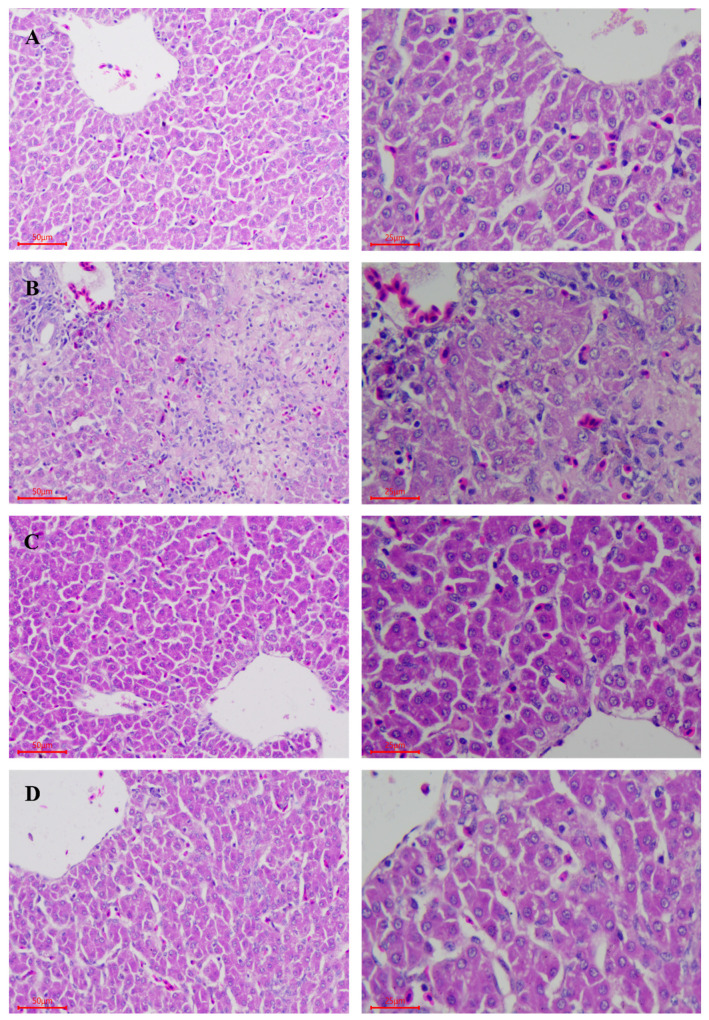
Histopathological changes of hepatocytes among groups. (HE, on the left, magnification 200×; on the right, magnification 400×). (**A**)CON group; (**B**) LPS group; (**C**) FA group; (**D**) FL group.

**Figure 3 toxins-14-00227-f003:**
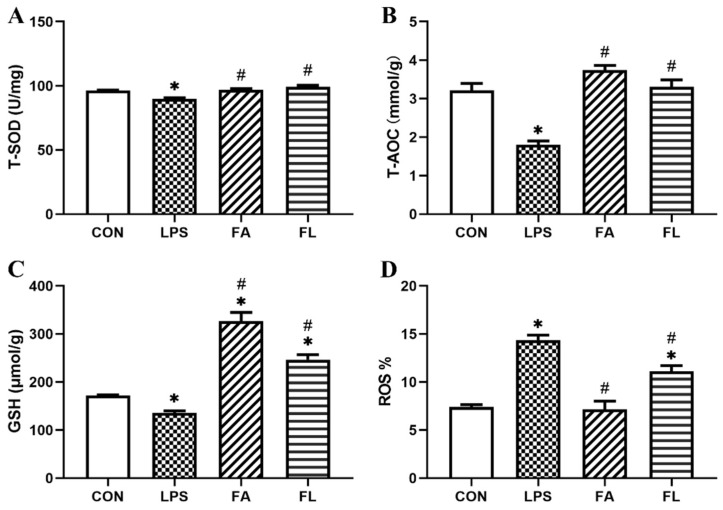
Activities of antioxidations among groups. The levels of (**A**) total superoxide dismutase (T-SOD, U/mg), (**B**) total antioxidant capacity (T-AOC, mmol/g), (**C**)glutathione (GSH, μmol/g), (**D**) reactive oxygen species (ROS, %). Values were shown in mean ± SE. * *p* < 0.05 compared with the CON group. # *p*< 0.05 compared with the LPS group.

**Figure 4 toxins-14-00227-f004:**
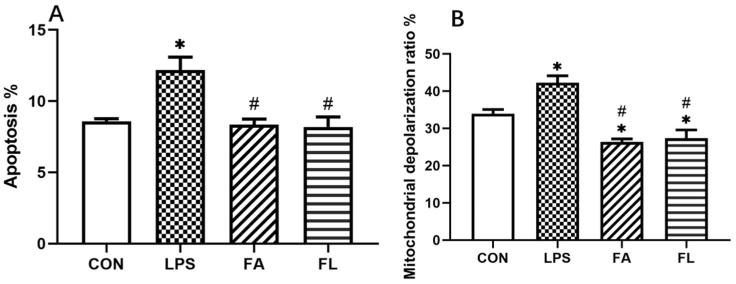
Apoptosis and changes in mitochondrial depolarization ratio. (**A**) apoptosis, % (**B**) mitochondrial depolarization ratio, %. Values were shown in mean ± SE. * *p* < 0.05 compared with the CON group. # *p* < 0.05 compared with the LPS group.

**Figure 5 toxins-14-00227-f005:**
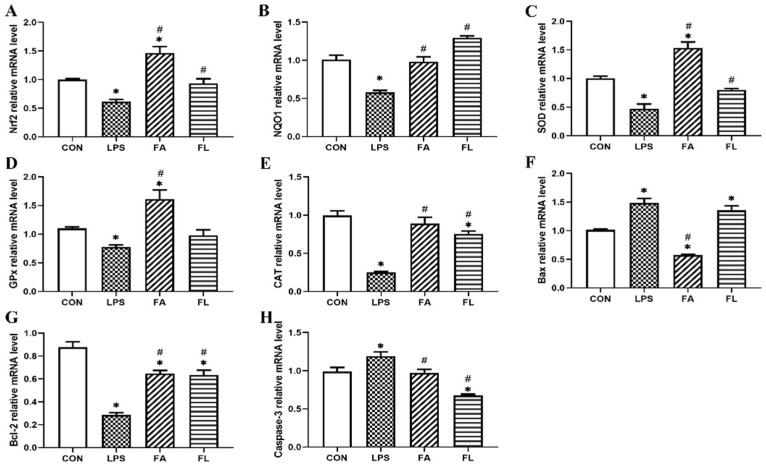
Expressions of antioxidant-related and apoptosis-related associated genes. The relative mRNA levels of (**A**) nuclear factor erythroid 2-related factor 2 (Nrf2); (**B**) NAD(P)H dehydrogenase quinone 1 (NQO1); (**C**) superoxide dismutase (SOD); (**D**) glutathione peroxidase (GSH-Px); (**E**) catalase (CAT); (**F**) Bcl-2-associated X protein (Bax); (**G**) B-cell lymphoma-2 (Bcl-2); (**H**) cysteine-aspartic acid protease (Caspase-3). Values were shown by mean ± SE. * *p* < 0.05 compared with the CON group. # *p* < 0.05 compared with the LPS group.

**Table 1 toxins-14-00227-t001:** Ingredient composition and calculated nutrient content of basal diet.

Item	Content (%)
Corn	59.50
Soybean meal	32.90
Vegetable oil	4.65
CaCO_3_	0.50
CaHPO_4_	1.60
NaCl	0.30
Choline	0.10
DL-Methionine	0.12
Premix ^1^	0.33
Calculation of nutrients	
Metabolizable energy (MJ kg^−1^)	12.80
Crude protein	19.70
Lysine	1.08
Methionime	0.40
Methionime + Cystine	0.74
Calcium	0.77
Nonphytate P	0.40

^1^ Provided per kg for diet: vitamin A (all-trans retinol acetate), 12,500 IU; cholecalciferol, 2500 IU; vitamin E (all-rac-a-tocopherol acetate), 18.75 IU; vitamin K (menadione Na bisulfate), 5.0 mg; thiamine (thiamine mononitrate), 2.5 mg; riboflavin, 7.5 mg; vitamin B6, 5.0 mg; vitamin B12, 0.0025 mg; pantothenate, 15 mg; niacin, 50 mg; folic acid, 1.25 mg; biotin, 0.12 mg; Cu (CuSO_4_·5H_2_O), 10 mg; Mn (MnSO_4_·H_2_O), 100 mg; Zn (ZnSO_4_·7H_2_O), 100 mg; Fe (FeSO_4_·7H_2_O), 100 mg; I (KI), 0.4 mg; and Se (Na_2_SeO_3_), 0.2 mg.

**Table 2 toxins-14-00227-t002:** Primers used in qRT-PCR.

Gene	Accession Number	Primer Sequence (5′-3′)	Product Length (bp)
*Nrf2*	XM_015289381.3	F: ACATCTTTTCTCATGATGGGTA R: TCACGAGCCCTGAAACCAA	82
*NQO1*	NM_001277621.1	F: TCTCTGACCTCTACGCCAT R: TCTCGTAGACAAAGCACTCGG	93
*SOD*	NM_205064	F: AGGGGGTCATCCACTTCC R: CCCATTTGTGTTGTCTCCAA	122
*GSH-Px*	NM001277853	F: TTGTAAACATCAGGGGCAAA R: ATGGGCCAAGATCTTTCTGTAA	164
*CAT*	NM_001031215.1	F: GGTTCGGTGGGGTTGTCTTT R: CACCAGTGGTCAAGGCATCT	211
*Bax*	XM_422067	F: TCCTCATCGCCATGCTCAT R: CCTTGGTCTGGAAGCAGAAGA	195
*Bcl-2*	NM_205339	F: TCCTCATCGCCATGCTCAT R: CCTTGGTCTGGAAGCAGAAGA	205
*Caspase-3*	XM_015276122.3	F: TGCTCCAGGCTACTACTCC R: CCACTCTGCGATTTACACGA	134
*β-actin*	NM_205518.1	F: AAGGATCTGTATGCCAACACA R: AGACAGAGTACTTGCGCTCA	148

*Nrf2*: nuclear factor erythroid 2-related factor 2, *NQO1*: NAD(P)H dehydrogenase quinone 1, *SOD*: superoxide dismutase, *GSH-Px*: glutathione peroxidase, *CAT*: catalase, *Bax*: B-cell lymphoma-2, *Bcl-2*: Bcl-2-associated X protein, *Caspase-3*: cysteine-aspartic acid protease-3, β-actin: Reference gene.

## Data Availability

The datasets used and/or analyzed during the current study are available from the corresponding author upon reasonable request.

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
