# Peer review of "Effects of Dietary Ferulic Acid Supplementation on Hepatic Injuries in Tianfu Broilers Challenged with Lipopolysaccharide"

_toxins, 2022, doi:10.3390/toxins14030227_

Round 1
Reviewer 1 Report
After examining the manuscript, we consider that there are aspects that need to be reviewed:
(i) In the abstract, sometimes they explain the acronyms (LPS, FA), but others like BW and FCR do not and it would be convenient if they explained them to better understand the abstract, because otherwise you have to go directly to look for it in the text .
iii) In general, the subsections of each section are missing. Also, I would suggest expanding the introduction a bit more. In addition, I have not seen the concentration of ferulic acid and/or LPS used...or if they have made several concentrations...these aspects must be perfectly specified in material and methods. The protocols must be described in terms of material and methods so that the article can be understood without the need to go to other references.
(iii) Aspects such as:
-Line 16: italicizes the p-value
-Line 21 (keywords): order them alphabetically.
-Line 23: they have not followed the guidelines of the template proposed by MDPI for this journal.
-Line 24: the same as the previous one. Please review all sections and follow the guidelines proposed by MDPI.
-Line 27: a space is missing before the reference
-Line 33: put in the same [] the reference 6 and 7, not separately
-Line 39 and 40: change keap1 are to keap1 is; change Nrf2 transports to to Nrf2 is transported to
-Line 47 and 48: they could put the abbreviation and its name in the same parentheses, that is, it should look like this: Ferulic acid (FA, 4-hydroxy....)
-Line 49: put seed in the plural and add to the species L., remaining as: Ferula foetida L.
-Line 93 (Table 1): explain what DL-Met is
-Line 111: change d to day
-Line 124: explain the protocol a little more even if you have used a commercial kit or specify the kit used and summarize your protocol.
-Line 128: explain the concentration of DCFA-DA used and its dissolution medium, as well as explain the procedure of said test. It is also convenient that they explain the controls that have been carried out (positive, negative) and the number of repetitions of each test.
-Line 167 (figure 1): it is advisable to put the meaning of the legend of the graphs at the bottom of the figure. Also, change the X axis and where you put Time, put Time (days) so that you don't always have to put the number accompanied by the d.
-Line 203 (Figure 4): an upper line is seen on the graphs as if they had been cut off
-Line 206: the subtitle is in bold. Put as a subsection and italicize
Author Response
Please see the attachment。

Reviewer 2 Report
The study (Effects of dietary Ferulic acid supplementation on hepatic injuries in Tianfu broilers challenged with lipopolysaccharide) is an interesting study but it needs some extra analysis. So, I will give Major revision
- There are very important analysis must be measured to complete the study such as Glutathione S-transferase (GST), Peroxidase and Catalase enzymes. Its very important enzyme in detoxification studies and H2O2 reduction
- I think the test ROS detection which made is general and authors must determine the most ROS famous radicals individuals such as H2O2, Superoxide anion radicals and Hydroxyl radicals
- There are also important non enzymatic antioxidants like vitamin E, vitamin C didnot measure
- There are some minor corrections in pdf file please made

Round 2
Reviewer 2 Report
I want to thank authors for there scientific reply. I am sure you can't determine every the antioxidant parameters even you don't have all facilities for that but I think if my suggest parameters done it will be very completed study. Anyway I am satisfied now and convinced by your improvements